# Progesterone Metabolism by Human and Rat Hepatic and Intestinal Tissue

**DOI:** 10.3390/pharmaceutics13101707

**Published:** 2021-10-16

**Authors:** Zoe Coombes, Katie Plant, Cristina Freire, Abdul W. Basit, Philip Butler, R. Steven Conlan, Deyarina Gonzalez

**Affiliations:** 1Reproductive Biology and Gynaecological Oncology Group, Swansea University Medical School, Singleton Park, Swansea SA2 8PP, UK; z.coombes@swansea.ac.uk (Z.C.); R.S.Conlan@swansea.ac.uk (R.S.C.); 2Cyprotex, No.24 Mereside, Alderley Park, Nether Alderley, Cheshire SK10 4TG, UK; K.Plant@cyprotex.com (K.P.); P.Butler@cyprotex.com (P.B.); 3Kuecept Limited, Potters Bar, Hertfordshire EN6 1TL, UK; Cristina.Freire@catalent.com; 4Department of Pharmaceutics, UCL School of Pharmacy, University College London, 29-39 Brunswick Square, London WC1N 1AX, UK; a.basit@ucl.ac.uk

**Keywords:** progesterone, intestinal metabolism, liver metabolism, aldo keto reductase, AKR, aldehyde oxidase, AOX

## Abstract

Following oral administration, the bioavailability of progesterone is low and highly variable. As a result, no clinically relevant, natural progesterone oral formulation is available. After oral delivery, first-pass metabolism initially occurs in the intestines; however, very little information on progesterone metabolism in this organ currently exists. The aim of this study is to investigate the contributions of liver and intestine to progesterone clearance. In the presence of NADPH, a rapid clearance of progesterone was observed in human and rat liver samples (t_1/2_ 2.7 and 2.72 min, respectively). The rate of progesterone depletion in intestine was statistically similar between rat and human (t_1/2_ 197.6 min in rat and 157.2 min in human). However, in the absence of NADPH, progesterone was depleted at a significantly lower rate in rat intestine compared to human. The roles of aldo keto reductases (AKR), xanthine oxidase (XAO) and aldehyde oxidase (AOX) in progesterone metabolism were also investigated. The rate of progesterone depletion was found to be significantly reduced by AKR1C, 1D1 and 1B1 in human liver and by AKR1B1 in human intestine. The inhibition of AOX also caused a significant reduction in progesterone degradation in human liver, whereas no change was observed in the presence of an XAO inhibitor. Understanding the kinetics of intestinal as well as liver metabolism is important for the future development of progesterone oral formulations. This novel information can inform decisions on the development of targeted formulations and help predict dosage regimens.

## 1. Introduction

Natural progesterone (Figure 1) is a highly permeable, poorly soluble steroid hormone which is found endogenously in both males and females. In females, progesterone maintains pregnancy and modulates the menstrual cycle; and in males, progesterone influences spermiogenesis, sperm capacitation/acrosome reaction and testosterone biosynthesis in the Leydig cells. In both genders, it exerts cerebroprotective effects and potentiates neural myelination [1]. Synthetic variants of progesterone are commonly prescribed for a range of female indications, such as contraception, hormone replacement therapy and certain hormone-responsive cancers [2]. However, when taken orally for release in the upper gastrointestinal (GI) tract, natural progesterone has very low bio-availability due to its poor absorption and extensive pre-systemic metabolism. This leads to considerable variability in the pharmacokinetics of progesterone among individuals [3]. Several research studies have been undertaken to develop an effective natural progesterone formulation [4,5,6]. Still, in the UK, natural progesterone is only prescribed as a vaginal suppository for luteal support as part of an assisted reproductive technology treatment in infertile women who are unable to use or tolerate other vaginal preparations.

The poor oral bioavailability of progesterone has primarily been attributed to extensive metabolism in the liver [7]. However, in the current literature, very little information exists on progesterone intestinal metabolism, although extensive enzymatic expression is found throughout the small intestinal mucosa [8,9,10,11,12]. Furthermore, the small intestine contains appreciable levels of bacteria [13,14]. The presence of bacteria is likely to further impact the stability of progesterone. Indeed, previous research within this group has demonstrated that progesterone undergoes a substantial amount of degradation in large intestinal contents [15]. It is, therefore, important to consider the metabolising abilities of the small intestine when preparing drugs for oral administration.

By examining the type of metabolism that occurs in liver and intestines, extensive research has shown that the cytochrome P450 (CYP450) family of enzymes is the main contributor to the inactivation of progesterone [16]. Although evidence also shows that progesterone is metabolised by aldo-keto reductases (AKR) [17,18], little is known about how different AKR subtypes affect total progesterone inactivation. In addition, a substantial amount of research into progesterone metabolism and bioavailability has been carried out in different animal species, particularly rats [19,20]. However, the metabolising abilities of many drugs differ between species often making animal models poor surrogates for humans [21]. For example, the bioavailability of several synthetic progestins has been shown to differ substantially between human and animal species [22], and many more such discrepancies have been documented [23]. It is thus important to compare animal and human models, as differences could have considerable implications for human risk.

In the absence of a clinically relevant progesterone oral formulation, it is important to understand the stability of progesterone when it enters the liver and intestines. More information is needed to guide future development. Therefore, during this study, the contributions of liver and intestine to progesterone metabolism will be assessed in both rats and humans. The roleof AKRs in this metabolism will also be investigated.

## 2. Materials and Methods

### 2.1. Materials

Progesterone (catalogue no. P0130) and positive controls, midazolam (catalogue no. 1443599), dolasetron (catalogue no. 1224959) and phthalazine (catalogue no. P38706) were purchased from Sigma Aldrich, Dorset, UK. NADPH tetrasodium salt (catalogue no. 10107824001) was bought from Roche Applied Science (Penzburg, Germany). Enzyme inhibitors Finasteride (catalogue no. 15472192), Flufenamic Acid (catalogue no. 165920500) and Menadione (catalogue no. 0210225925) were purchased from Fisher Scientific (Loughborough, UK). Epalrestat (catalogue no. SML0527), Diazepam (catalogue no. D0899) and Allopurinol (catalogue no. A8003) were purchased from Merck Millipore (Dorset, UK). Metoprolol (catalogue no. PHR1076), Tween 80 (catalogue no. P1754) and DMSO (catalogue no. D2650) were purchased from Sigma Aldrich. Rat intestinal and rat liver homogenates were purchased from Biopredic international (Saint Grégoire, France) and arrived pre-prepared at pH 7.4. These were obtained from a pool of male Sprague Dawley rats. Pooled samples of human intestines were purchased from In Vitro technologies and were obtained from 3 male and 3 female donors. Human liver homogenate was purchased from UK Human Tissue Bank (Leicester, England) and contained pooled samples from 9 male and 3 female donors. Pooled male and female human liver and intestinal cytosol were purchased from In Vitro technologies. All other chemicals and solvents were of HPLC reagent grade or the highest commercially available grade.

### 2.2. Tissue Homogenate Stability Assay

Progesterone stability was determined in rat and human intestinal mucosa and liver homogenates. In addition, progesterone was incubated in stock solution containing phosphate-buffered saline, 0.1% Tween80 and 1% DMSO. The solution was maintained at pH 7.4, and progesterone was shown to be stable for up to 6 h at a concentration of 1 µM. Stability assays were carried out either in the presence or absence of 1mM of the cofactor NADPH. The physiological concentration of NADPH was approximately 100 µM [26]. However, for the purpose of this research, a concentration of 1 mM was used to ensure an excess of co-factor was available and not rate-limited throughout the incubation period.

The protein concentration for the rat and human liver and intestinal homogenates was adjusted to 5 mg/mL. Next, tissue homogenates were incubated for 5 min in a Thermomixer Comfort by Eppendorf in a 1 mL deepwell RNAse/DNAse 96-well plate (Thermo Fisher Scientific) until reaching a temperature of 37 °C. A final concentration of 1 µM progesterone and 1 mM NADPH (where required) was added to the homogenate, and samples were immediately taken (t–0). Next, the homogenate solution was incubated with shaking (700 rpm) at 37 °C, and further samples were taken at timed intervals. To stop the enzymatic reaction, samples were immediately diluted in ice-cold acetonitrile at a ratio of 1:3 sample:acetonitrile. The samples were then centrifuged at 3000 rpm for 30 min to pellet the homogenates. Following centrifugation, the supernatant was removed and diluted 1:1 with milli-Q water. Finally, metoprolol was added to the solution as an internal standard for LC-MS/MS at a concentration of 0.2 µM.

### 2.3. Human Cytosol Stability Assay

Progesterone stability was assessed in intestinal and liver cytosol in the presence or absence of NADPH cofactor and specific enzyme inhibitors. For both intestinal and liver cytosol, the protein concentration was adjusted to 0.5 mg/mL. Both human liver and intestinal cytosol (with 0.1 mM EDTA) were then pre-incubated for 10 min in a thermomixer comfort by Eppendorf in an RNAse/DNAse 96-well plate until reaching a temperature of 37 °C. Next, a final concentration of 100 µM of enzyme inhibitors were added to the cytosol. The concentration of 100 µM was selected to ensure substantial inhibitors were present for successful inhibition of in vitro metabolism. The enzyme inhibitors used during this study were: Epalrestat (AKR1B1 inhibitor), Diazepam (AKR1C inhibitor), Finasteride (AKR1D1 inhibitor), Diazepam + Flufenamic Acid (AKR1C inhibitor cocktail), Menadione (aldehyde oxidase inhibitor) and Allopurinol (xanthine oxidase inhibitor).

Next, the mixtures were incubated for an additional 5 min, and NADPH was added at a final concentration of 1 mM. Finally, progesterone (1 µM) was added and mixed by pipetting to initiate the reaction. Immediately following reaction initiation, a 50 µL sample was taken. The sample was then diluted in 100 µL of acetonitrile to terminate enzymatic reaction. Samples were centrifuged at 4 °C and 2500 rpm for 30 min. Following centrifugation, the supernatant was removed and diluted 1:1 with milli-Q water. Metoprolol was added to the solution as an internal standard for LC-MS/MS at a concentration of 0.2 µM. This process was followed for sampling at all required time intervals.

### 2.4. Sample Analysis

Liquid chromatography with tandem mass spectrometry (LC MS/MS) was used to analyse samples obtained from both cytosol and homogenate stability assays (see Appendix A Figure A1 for example spectra). The LC MS/MS system used for this study consisted of an Acquity Binary Solvent Manager (BSM), 2777 Ultra-high-pressure autosampler, Acquity 4-position heated column manager and a Xevo-TQ MS Triple Quadrupole mass spectrometer (Water Ltd., Herts, UK). The column used was Aquity HSS T3 (1.8 µm) 2.1 × 50 mm (Water Ltd., Herts, UK), and the samples were run with an injection volume of 8 µL at 70 °C. A solvent system and gradient with a flow rate of 600 µL/minute was used to perform this analysis. The mobile phase consisted of 10 mM ammonium formate with 0.1% *v/v* formic acid in water (A) and acetonitrile (B). The following gradient conditions were used: 0.0–1.0 min, 100% A; 1.0–1.4 min, 5% A; 1.4–1.8 min, 100% A.

### 2.5. Data Analysis

Following LC-MS/MS, TargetLynx (Waters) was used to determine the peak area of progesterone and metoprolol internal standard using auto-integration. Next, progesterone and control substrate (metoprolol, phthalazine and dolasetron) stability was determined by taking the average peak area ratio of three replicates. The percent of substrate present at each timepoint relative to the 0 min sample was calculated from the LC-MS/MS peak area ratios (compound peak area/internal standard peak area). Graphs were created using Microsoft Excel, and statistical analysis was carried out using Excel extension XLSTAT 2021.1. The Kolmogorov–Smirnov test was used to assess normality followed by Student’s *t*-test. Results were considered statistically significant when *p* < 0.05. Graphpad was used to calculate the half-life (t_1/2_) using the equation t_1/2_ (min) = 0.693/k where k is elimination rate constant.

## 3. Results and Discussion

First-pass metabolism and GI tract variability are common causes of variable and incomplete bioavailability for orally dosed drugs [27]. Understanding the kinetics of progesterone depletion during first-pass metabolism in the liver and intestines is, therefore, important to consider whilst developing new oral formulations. Therefore, the stability of progesterone was assessed in intestinal and hepatic homogenates from rat and human, as well as in human cytosol. The degradation of progesterone within these samples can be seen in Figure 2, Figure 3 and Figure 4, including in the presence of AKR inhibitors. The half-life of progesterone under different conditions can be seen in Table 1. Progesterone remained 99% detectable after 2 h in control media, confirming that decreases in the hormone concentration were due to enzymatic activity within the samples (Figure 2a). The internal standard, Motropolol, has been previously shown to be stable in solution for at least 20 h in PBS solution [28]. Positive controls, which were also incubated in human intestinal homogenates and cytosol, showed substantial enzymatic activity, further validating the methodology (Appendix A Figure A2).

Progesterone Clearance in Liver (Figure 2a) and Intestinal (Figure 2b) Homogenates.

Progesterone Clearance in Human Liver (Figure 3a) and Intestinal (Figure 3b) Cytosol +/− AKR inhibitors.

Progesterone Clearance in Human Liver Cytosol.

### 3.1. Progesterone Metabolism by Human and Rat Liver and Intestinal Homogenates

In this study, the rate of progesterone metabolism in the liver and intestine of rat and human species was determined. Rat and human liver homogenates showed a similar, extensive metabolism of progesterone, whereby progesterone was degraded to less than 2% of its original concentration within 10 min of incubation in the presence of NADPH cofactor. In line with previous research [29], in the absence of NADPH, the depletion rate of progesterone was reduced, with over 40% of the drug remaining in both species after two hours of incubation (Figure 2a). The half-life in these conditions was calculated as just 2.7 min for both rat and human samples (Table 1). No significant differences were observed in the rate of degradation between the two species. In rat and human intestinal homogenates, including a combination of duodenal, jejunal and ileal mucosa, progesterone was degraded to 77% and 74% in 60 min, respectively, in the presence of NADPH (Figure 2b). In the absence of NADPH over 90% of progesterone was detectable in rat intestine after 60 min. However, in human intestinal samples, only 76% of progesterone was detectable by LC MS/MS, a difference that was deemed significant by *t*-test (*p* < 0.05) (Figure 2b).

In vivo, progesterone can be metabolized to over 30 metabolites [30], with 5α- and 5β-pregnanolone being most common. Progesterone is also converted to 11-deoxycorticosterone (a potent mineralocorticoid), 20-dihydroprogesterone (with weak progestogenic activity), and 17α-hydroxyprogesterone (inactive) [30]. It is likely that these metabolites are produced through oxidation by enzymes such as CYP450 that are highly expressed in the liver and intestinal wall, which is a likely mode of metabolism seen within this study [24]. The differences seen in rat and human intestinal homogenate in the absence of NADPH suggests a difference in enzymatic expression within the gut wall of rats and humans. In particular, the presence of non-NADPH dependent enzymes appears to be more prominent in the human intestinal wall. Indeed, previous research has shown that rodent animals and humans have a variety of different metabolizing enzymes in the small intestine [31]. Moreover, recent publications have highlighted the importance of selecting appropriate animal models to simulate human physiology in preclinical studies [21]. The data suggest that rat samples may be a poor choice of model for testing future progesterone oral formulations; therefore, for the remainder of this research, only human samples will be utilised.

### 3.2. Progesterone Metabolism in Liver and Intestinal Cytosol and the Role of Aldo-Keto Reductases

The CYP450 metabolism is commonly described as occurring in the liver and gut wall [9,10]. The CYP-3A subfamily of enzymes, for example, includes known metabolizers of progesterone that are expressed in the GI tract at rates that may even exceed that found in the liver [11]. This has emphasised the CYP family of enzymes when investigating the first-pass metabolism of progesterone [16]. However, AKRs have been shown to extensively metabolise progesterone and are present in both liver and intestine [17,18]. AKR1C, in particular, is a known metaboliser of progesterone. In fact, it serves as one of the three major contributors to hepatic depletion of progesterone in lactating dairy cows by acting as 3-, 17- and 20-ketosteroid reductases to varying degrees [32]. Despite the available research on the role of AKRs in progesterone metabolism, a paucity of information exists regarding the relative contributions of the different AKR subtypes to total progesterone inactivation. Furthermore, the rate and extent of progesterone depletion in the intestine in the presence of AKR inhibitors is currently absent from the literature. Therefore, the data presented herein aim to shed light on the effect of the different AKRs on progesterone depletion in liver and intestinal samples. Note that, as AKRs are dependent of NADPH as a cofactor, all the following experiments were carried out in the presence of NADPH. As a control, the samples were also incubated in liver cytosol with AKR inhibitors in the absence of NADPH, but no differences were observed (Appendix A Figure A3).

These data suggest that AKR1C plays an important role in progesterone depletion in the human liver, with AKR1C3 being the most likely primary subtype involved in this reaction. The rate of progesterone depletion was significantly reduced in the liver in the presence of anAKR1C inhibitor cocktail (*p* < 0.01) with a half-life of 35.6 min, compared to 13 min in the absence of an inhibitor. Interestingly, no significant difference was observed in the presence of the AKR1C1 inhibitor, diazepam (Figure 3a). As well as diazepam, the AKR1C inhibitor cocktail contains flufenamic acid, which has a much higher potency against the AKR1C3 isoform [33], suggesting that AKR1C3 is a stronger metaboliser of progesterone than AKR1C1. However, in intestinal samples, although a slight reduction can be seen in the rate of progesterone depletion (Figure 3b), neither the AKR1C1 inhibitor nor the AKR1C inhibitor cocktail had any significant effect on the rate of progesterone depletion. A publication by Pratt-Hyatt et al. shows that certain subtypes of AKR1C are more highly expressed in the intestine than in the liver, which may explain the differences seen within this study [34]. The rate of progesterone depletion was also found to be significantly reduced by inhibitors of the 5β-steroid reductase AKR1D1 (*p* < 0.05) (Figure 3a). The 5β-pathway is a main metabolic pathway for progesterone metabolism [17]; thus, inhibition of this pathway likely slowed the inactivation of progesterone in this instance. This effect was not seen in intestinal samples, however, in which the AKR1D1 inhibitor did not cause a significant reduction in the rate of progesterone depletion. Conversely, AKR1B1 inhibition caused a significant reduction in the rate of progesterone depletion in the intestines (Figure 3b, *p* < 0.01) as well as in the liver. Progesterone’s half-life significantly increased from 13 min to 18 min in liver, and from 129.5 min to 184.1 in intestines (Table 1).

Together this data elucidates the roles of the different AKR subtypes in progesterone metabolism. Such information may be useful when considering the development of new progesterone oral formulations. It is also worth noting the differential expression of AKRs between genders [35], which may further impact successful formulation delivery. Of course, many important differences within human samples, including sex differences and disease state, should also be considered [36,37].

### 3.3. Progesterone Metabolism by Non-NADPH Dependent Enzymes

Despite progesterone depletion being significantly reduced in the absence of NADPH in human liver, sufficient enzymatic activity was observed (Figure 2a), suggesting an important role for NADPH-independent metabolising enzymes in progesterone depletion in this organ. There is currently very little information on progesterone metabolism by enzymes that does not require NADPH as an electron donor. Two important NADPH-independent enzymes that may contribute to progesterone metabolism are aldehyde oxidase (AOX) and xanthine oxidase (XAO). Both AOX and XAO are cytoplasmic enzymes that catalyse the biotransformation of many drugs, with the highest levels being found in the liver and intestine [38]. AOX has gained considerable interest in recent years because of examples of its role in the rapid clearance of drug compounds in development [39].

Progesterone was incubated in liver cytosol with allopurinol, an XAO inhibitor which has been reported to cause inhibition of progesterone production in luteal cells [40,41]. However, no direct interaction of XAO and progesterone has been reported. After 2 h of incubation in liver cytosol, no difference was observed in the rate of progesterone degradation in the presence and absence of allopurinol (Figure 4a), suggesting that there is no significant interaction between XAO and progesterone metabolism in the liver. Conversely, the rate of progesterone metabolism was significantly reduced in the presence of the AOX inhibitor, menadione. As well as inhibiting AOX activity [42], menadione also inhibits several NADPH-dependent enzymes, including aniline-p-hydroxylase. It can also negatively affect CYP reductase activity [43]. Therefore, progesterone was incubated with menadione in the presence and absence of NADPH (Figure 4b). Menadione significantly inhibited progesterone degradation in the absence of NADPH (*p* = 0.002). Although a reduction in the rate of progesterone depletion was observed in the presence of NADPH, this was not deemed significant by *t*-test (Figure 4b), suggesting that the reduction in progesterone metabolism was primarily due to the inhibition of AOX activity. To further confirm the role of AOX in progesterone depletion, progesterone was also incubated in liver cytosol with AOX inhibitors hydralazine and chlorpromazine (Appendix A Figure A4). Both inhibitors resulted in a similar reduction in the rate of progesterone depletion, suggesting that AOXs play a previously unknown role in progesterone metabolism.

A limitation of this study is that it was carried out in pooled male and female samples. Sex differences in enzymatic expression are an important consideration in drug development, as hepatic and intestinal enzymes have been shown to differ between the sexes (Kennedy 2008). Nonetheless, this study provides an insight into how progesterone is degraded by healthy tissue. In future work, the degradation of progesterone should be assessed in separate male and female samples.

## 4. Conclusions

This study looked at the rate of progesterone depletion in human and rat liver, and in intestinal preparations. Rat and human were found to metabolise progesterone at a similar rate in both liver and intestine in the presence of an NADPH cofactor. In the absence of a cofactor, significant differences were seen between rat and human intestines, though no differences were observed in the liver under these conditions. In addition, AKR inhibitors of AKR1C, 1D1 and 1B1 were found to significantly reduce the rate of progesterone depletion in human liver. The AKR1B1 inhibitor also caused a significant reduction in the rate of progesterone depletion in human intestines. Although AKR1C and 1D1 reduced the rate of progesterone depletion, this was not found to be statistically significant by *t*-test. Furthermore, the inhibition of aldehyde oxidase caused a significant reduction in progesterone degradation in human liver. Together, these findings provide novel information on the contributions of liver and intestine to progesterone metabolism. Such information could help guide the future development of progesterone formulations and predict dosage regimens. Furthermore, this research also contributes to a more comprehensive knowledge of the safety and efficacy of progesterone. Although it is well known that this agent is rapidly converted to metabolites in the liver, this study demonstrates that the intestines also play a significant role in progesterone metabolism. Hence, the intestinal metabolism of progesterone should be taken into account when progesterone is prescribed to patients in conjunction with other drugs that may affect the function of the intestines, so the bioavailability of progesterone is not compromised.

## Figures and Tables

**Figure 1 pharmaceutics-13-01707-f001:**
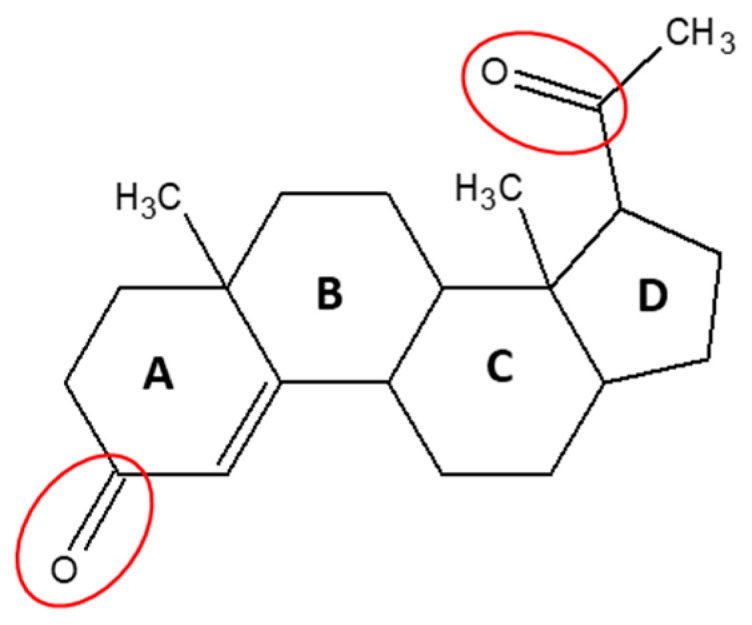
Chemical Structure of Progesterone. Reduction and hydroxylation of the two ketone groups (circled red) make progesterone vulnerable to liver and intestinal metabolism [24,25].

**Figure 2 pharmaceutics-13-01707-f002:**
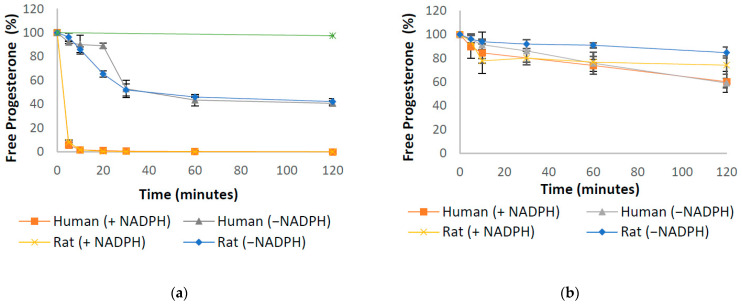
Stability profile of progesterone in human and rat liver (**a**) and intestinal (**b**) homogenates in the presence and absence of NADPH (*n* = 3, error bars are +/− SD).

**Figure 3 pharmaceutics-13-01707-f003:**
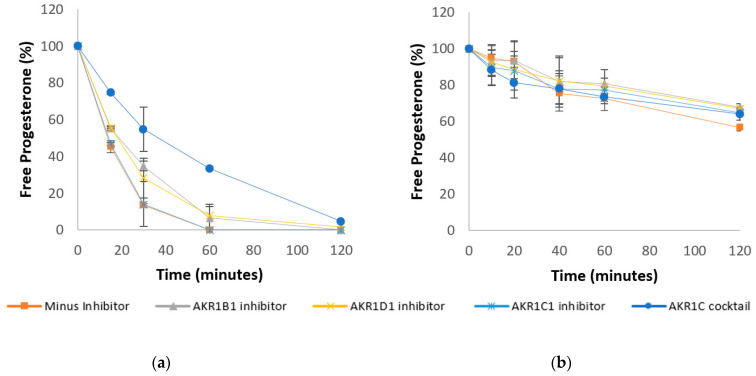
Stability profile of progesterone in human liver cytosol (**a**) and human intestinal cytosol (**b**) in the presence of 1mM of NADPH alone or with the addition of enzyme inhibitors (error bars are +/− SD).

**Figure 4 pharmaceutics-13-01707-f004:**
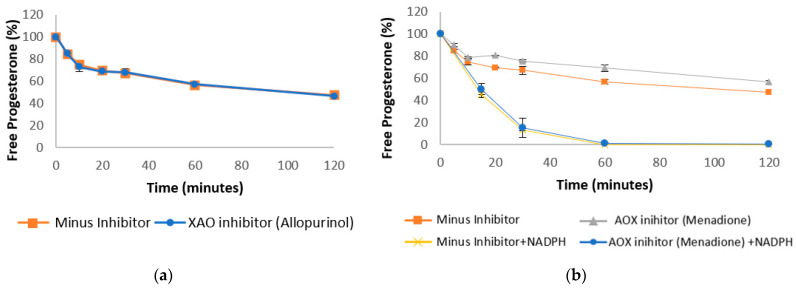
Stability profile of progesterone in human liver cytosol in the absence of NADPH (unless stated) and with the addition of XAO inhibitor (**a**) AOX inhibitor (**b**) (error bars are +/− SD).

**Table 1 pharmaceutics-13-01707-t001:** Half-life (t_1/2_) and concentration of progesterone after 60 min of incubations with named solution.

Solution	t_1/2_ (min)	% Drug Remaining After 60 min
Progesterone in liver homogenates (Figure 2a)		
Human liver homogenate + NADPH	2.7 ± 0.05	0.27 ± 0.26
Human liver homogenate	38.95 ± 11.68	43.32 ± 4.84
Rat liver homogenate + NADPH	2.72 ± 0.05	0.03 ± 0.0005
Rat liver homogenate	39.77 ± 10.16	46.04 ± 0.69
Progesterone in intestinal homogenates (Figure 2b)		
Human intestinal homogenate + NADPH	157.22 ± 27.41	73.92 ± 5.11
Human intestinal homogenate	155.84 ± 13.75	75.82 ± 9.35
Rat intestinal homogenate + NADPH	197.63 ± 2.91	76.67 ± 4.72
Rat intestinal homogenate	352.06 ± 36.87	90.93 ± 2.19
Progesterone in liver cytosol + NADPH (Figure 3a)		
Human liver Cytosol	13.73 ± 0.81	0.04 ± 0.01
Human liver Cytosol + AKR1B1 inhibitor	18.81 ± 0.96	6.6 ± 6.24
Human liver Cytosol + AKR1D1 inhibitor	18.46 ± 1.7	7.85 ± 6.31
Human liver Cytosol + AKR1C1 inhibitor	14.08 ± 0.24	0.11 ± 0.04
Human liver Cytosol + AKR1C1 cocktail	35.56 ± 6.67	33.5 ± 1.72
Progesterone in intestinal cytosol + NADPH (Figure 3b)		
Human intestinal Cytosol	129.52 ± 2.75	72.51 ± 6.48
Human intestinal Cytosol + AKR1B inhibitor	184.12 ± 18.95	80.87 ± 7.84
Human intestinal Cytosol + AKR1D1 inhibitor	181.74 ± 4.39	79.56 ± 4.44
Human intestinal Cytosol + AKR1C1 inhibitor	167.39 ± 3.03	77.38 ± 6.37
Human intestinal Cytosol + AKR1C cocktail	161.82 ± 14.46	73.56 ± 3.69
Progesterone in liver cytosol (Figure 4a)		
Human intestinal Cytosol	103.1 ± 9.38	56.6 ± 2.07
Human liver Cytosol + XAO inhibitor	99.63 ± 9.14	57.09 ± 2.01
Progesterone in liver cytosol (Figure 4b)		
Human intestinal Cytosol	103.1 ± 9.38	56.6 ± 2.07
Human liver Cytosol + AOX inhibitor	132.6 ± 4.69	69.14 ± 2.91
Human liver Cytosol + NADPH	13.73 ± 0.81	0.04 ± 0.01
Human liver Cytosol + NADPH + AOX inhibitor	15.49 ± 2.03	0.97 ± 0.88

## Data Availability

The data presented in this study are available on request from the corresponding author. Result can also be accessed at https://cronfa.swan.ac.uk/Record/cronfa42514.

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
