# Peer review of "Progesterone Metabolism by Human and Rat Hepatic and Intestinal Tissue"

_pharmaceutics, 2021, doi:10.3390/pharmaceutics13101707_

Round 1

Reviewer 1 Report

Coombes et al provides an insightful study on progesterone metabolism in both intestinal and hepatic tissues. The paper is well written and interesting to read. There are a few comments that I think may improve the paper.

  1. Could you provide the cat numbers of all the materials used, including the inhibitors.
  2. Were any desalting or SPE clean up methods used for these samples prior to LC-MS? Is so could a description be included.
  3. A description outlining why the dosages of NADPH (1 mM) and inhibitors (1oo uM) were chosen, is 100 uM physiologically relevant for all the inhibitors used. 
  4. Minor comment: the spacing between numbers and units is inconsistent eg lines 130 and 132
  5. Could a sample spectra be included in the appendix
  6. The figures should be presented in Graphpad is the analysis was performed with this software.      

Reviewer 2 Report

I congratulate the Authors on an interesting publication. I propose some corrections and additions to increase the value of the manuscript.

  1. What software used to calculate statistics. What test was used to check the distribution of variables?
  2. Minor errors, eg no space between the number and the value 8μl (make it uniform), incorrect entries: at 4oC, 37oC; (t1 / 2) ½ should be subscript.
  3. What was the short-term and long-term stability in room temperature and in the freezer for metoprolol?
  4. Parts are missing: discussion, after the results there is a summary.
  5. What are the limitations of the study?
  6. How can we use the obtained results in clinical practice?
